# Spatial Correlation and Influencing Factors of Environmental Regulation Intensity in China

**Lili Feng [1], Jingchen Shao [1], Lin Wang [2,3] and Wenjun Zhou [1,4,\*]**

[1] School of Management, Hebei GEO University, Shijiazhuang 050031, China; fenglili@hgu.edu.cn (L.F.); alice_121@163.com (J.S.)

[2] School of Ethnology, Northeastern University at Qinhuangdao, Qinhuangdao 066000, China; wl742088670@gmail.com

[3] Department of History, University of Punjab, Lahore P.O. Box 54590, Pakistan

[4] Research Center of Natural Resources Assets, Hebei GEO University, Shijiazhuang 050031, China

[\*] Correspondence: zhouwenjun@hgu.edu.cn

**Abstract:** In this study, we examined the spatial difference of environmental regulation intensity in 30 provinces (autonomous regions and municipalities directly under the central government) of China. It was found that there were significant differences in environmental regulation intensity in the four regions, with a decreasing trend of "west–central–northeast–east" on the whole. Applying the Theil index showed that intra-regional differences accounted for more than 85% of the overall differences in environmental regulation intensity. Goble Moran's I index was used to verify the spatial correlation of China's environmental regulation. It was found that the *p*-value of Goble Moran's I index was less than 10% in 7 years from 2010 to 2019. It was verified that the environmental regulation intensity in China has had a spatial correlation. In addition, a positive spatial correlation between the environmental regulation intensity in each province was found, indicating that an increase in the environmental regulation intensity of one province will lead to an increase in the intensity of environmental regulation in neighboring provinces. Finally, through the construction of a spatial Markov model to test the spillover effect of environmental regulation intensity in China, it was found that the local environmental regulation intensity will change to different degrees when there are spatial differences in the intensity of environmental regulation in neighboring provinces. This research will be helpful for provincial governments to formulate appropriate environmental regulation targets based on regional characteristics, which is of great significance for China's and other countries' green economic development and other countries to solve the contradiction between environmental pollution and economic development.

**Keywords:** environmental regulation intensity; spatial correlation; spillover effect

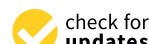



## 1. Introduction

Realizing a balance between environmental protection and economic development and achieving a win–win situation between environmental quality improvement and high-quality economic development are urgent problems to be solved in China. China's economy has made remarkable progress in the past 40 years, but the long-term extensive economic development model has increased the burden on the ecological environment. According to the 2020 Global Environmental Performance Index Report (EPI) (The website of the EPI report is https://epi.yale.edu/) jointly released by the Yale University and other research institutes on 20 March 2022, China ranked 120th out of 180 countries with an environmental performance index score of 37.3. This reflects that China's environmental improvement lags behind the global average. Refusing the idea that 'environmental governance and economic development can only choose one or the other' is an important part of the construction of a socialist economic and ecological civilization with Chinese characteristics, and it is also a great challenge facing the Chinese government.

However, it is difficult to effectively solve the dilemma between environmental quality improvement and high-quality economic development by the market itself. In the real-world market economy, environmental protection and economic development are often contradictory, and economic growth itself cannot effectively solve the problem of environmental degradation. In addition, due to the negative externalities of environmental problems caused by the nature of public goods and environmental resources, the existence of micro-economic subject opportunism, and China's long-term economic growth mode with high energy consumption, non-green production in the free market economy has a first-mover advantage and enterprises have insufficient motivation to independently carry out green production. Therefore, encouraging enterprises to produce green products by market mechanism alone is difficult, and the market failure problem needs to be solved by government intervention.

In order to achieve the goal of environmental protection, the government or the environmental protection agency, will implement a series of policies, laws and regulations, measures and means in the form of injunction or market incentives in order to force the use of environmental resources of each subject in their activities to undertake the responsibility of protecting the environment and natural resources, as well as realize the sustainable development of economic and social issues. These policies, regulations, measures and means are called environmental regulations.

Environmental regulation has become an important means to solve the problems of environmental pollution and ecological destruction in China. As an important means of environmental governance, environmental regulation is an important starting point for China to fight the battle of pollution prevention and establish an environmental governance system. It is also the key to ensure the coordination of economic development and environmental governance. Provincial governments have also established a large number of local regulations to promote the improvement of local environmental quality. However, due to the great differences in resource endowment, industrial structure, urban and rural structures, technological progress, and other aspects, the degrees and characteristics of environmental pollution in different regions are also different. Therefore, provincial governments formulate and implement environmental regulation policies according to local economic and environmental pollution conditions. As a result, the intensity of environmental regulations made by different provincial and municipal governments varies to different degrees. Under the environmental protection tax implemented in 2018, for example, the environmental protection tax rates formulated by provinces (regions and municipalities directly under the central government) are quite different, as shown in Table 1.

It can be seen that there are great differences in the intensity of environmental regulations in different provinces, so the formulation of environmental regulations should be closely related to geospatial factors. Therefore, on the basis of analyzing the spatial differences of environmental regulation intensity in different regions, scholars need to clarify the influence of changes in environmental regulation intensity in neighboring provinces on the formulation of environmental regulation policies in local provinces in order to guide the local government to formulate and implement environmental regulation policies in accordance with local conditions so as to maximize the effectiveness of environmental regulation policies and improve local ecological environment quality.

Accordingly, the authors of this this paper established a theoretical framework based on "spatial heterogeneity–spatial correlation–spatial spillover effect". Focused on the intensity of environmental regulation itself, this theoretical framework was used to investigate the definition of its spatial effect and to deeply analyze the theoretical and action mechanisms related to this spatial effect of environmental regulation, providing theoretical and logical bases for subsequent research. In order to enrich the research framework of the spatial evolution process and the multi-factor driving mechanism of environmental regulation, researchers should fully explore and study the spatial effects of environmental regulation and discuss the regional differences, dynamic evolution rules of spatial patterns, and influencing factors of environmental regulation in China. It is helpful for the govern-

ment to further adopt appropriate means of environmental regulation and has important reference value and significance for promoting the coordinated development of economic growth and environmental governance.

**Table 1.** Statistical table of environmental protection tax rate of China's provinces (%).

| Province | Water Pollutants | Tax Brackets | Province | Air Pollutants | Tax Brackets |
|---|---|---|---|---|---|
| Ningxia, Xinjiang, Gansu, Qinghai, Shanxi, Jilin, Liaoning, Shandong, Yunnan, Jiangxi, Zhejiang, Hubei, Tianjin, and Anhui | 1.4 | Low Level | Ningxia, Xinjiang, Gansu, Qinghai, Shanxi, Jilin, Liaoning, Shandong, Yunnan, Jiangxi, Fujian, Zhejiang, Tianjin, Anhui | 1.2 | Low Level |
| Fujian, Heilongjiang, and Shanxi | 2.1 | Middle Level | Guangxi, Guangdong, Heilongjiang, and Shanxi | 1.8 | |
| Guangxi, Sichuan, Guizhou, Hainan, and Guangdong | 2.8 | | Guizhou, Hainan, and Hunan | 2.4 | Middle Level |
| Chongqing and Hunan | 3 | | Hubei | 2.8 | |
| | | | Chongqing | 3.5 | |
| | | | Sichuan | 3.9 | |
| Shanghai | 4.8 | High Level | Henan, Jiangsu, and The Third grade of Hebei | 4.8 | High Level |
| Henan, Jiangsu, and The Third grade of Hebei | 5.6 | | The Second grade of Hebei | 6 | |
| The Second grade of Hebei | 7 | | Shanghai | 7.6 | |
| The First grade of Hebei | 11.2 | | The First grade of Hebei | 9.6 | |
| Beijing | 14 | | Beijing | 12 | |

## 2. Literature Review

In this paper, we summarize and expound relevant research contents from four aspects: the concept of environmental regulation, the quantification of environmental regulation, the spatial correlation of environmental regulation, and the factors affecting the spatial relevance of environmental regulation, as shown in Figure 1.

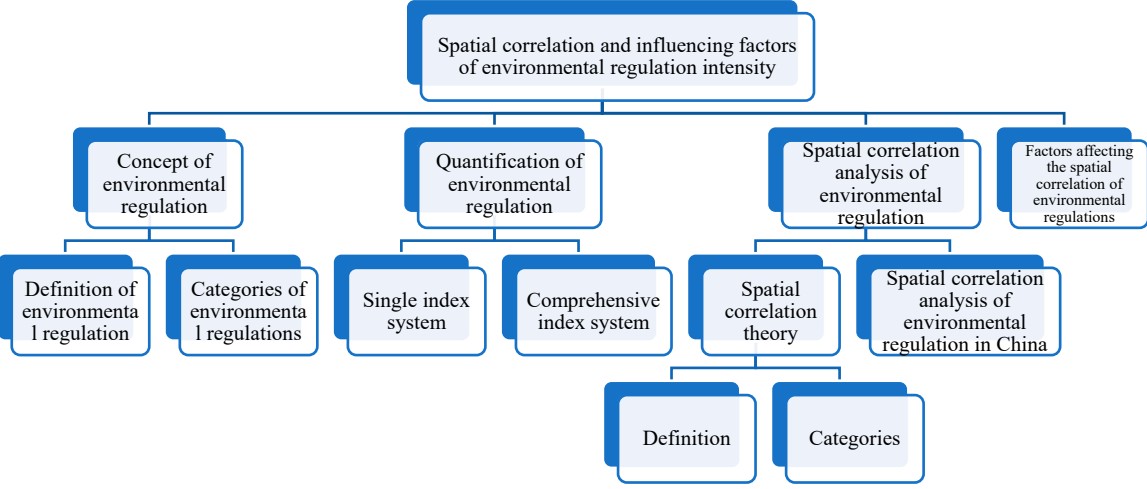

**Figure 1.** Literature review research diagram.

### 2.1. Research on the Concept of Environmental Regulation

Environmental regulation is the means by which the government restrains and restricts the activities of economic subjects in order to protect the ecological environment and realize the sustainable economic development of society. Jaffe et al. believed that environmental

regulation is a means implemented by government departments to intervene in adverse environmental effects [1]. Frondel et al. pointed out that environmental regulation is a tool of government governance policy and the main driving force of green technology innovation [2]. Wang et al. believed that environmental regulation is a traditional tool to solve environmental problems with the help of government forces, as well as an important means to improve enterprises' economic performance and environmental performance [3].

In terms of the recognition of the attribute of environmental regulation, regulation economics divides regulation into social and economic regulation. Social regulation aims at promoting social welfare, including ensuring people's safety, preventing public hazards and disasters, and protecting the environment. Economic regulation refers to the regulation of industries with natural monopolies and information asymmetry in order to prevent inefficient resource allocation and ensure fairness, including the establishment of import and export barriers, price restrictions, and production or quality restrictions. Zhang et al. stated that environmental regulation belongs to social regulation [4]. However, since environmental regulation involves economic regulation measures such as industry entry and exit barriers, Zhao reported that environmental regulation has both social and economic attributes [5]. The above analysis shows that the connotation of environmental regulation has undergone profound changes. In terms of the current theoretical and practical cognition, the connotations of environmental regulation include the following. Firstly, the subject of environmental regulation is not only the government but also enterprises, non-governmental organizations, and the public. Secondly, the means or tools of environmental regulation include both social regulation tools and economic regulation tools. The purpose of environmental regulation includes both social regulation purposes, such as environmental protection, and economic regulation purposes, such as the optimal allocation of natural resources. Therefore, environmental regulation is a regulation with both attributes. Thirdly, the object of environmental regulation is economic activity that will cause environmental damage. Fourthly, environmental regulation is a kind of behavior constraint that does not necessarily need a coercive force to enforce.

Environmental regulations are classified as command-and-control, market-incentive, and voluntary participation based on the different ways in which they produce binding effects [6]. The command-and-control type of environmental regulation refers to the control of enterprises' environmental behaviors, usually in the form of regulations and laws with penalties for violations, in order to achieve the environmental goals of the regulator, and most rules, regulations, and environmental-related standards are of this type. The market-incentive type refers to a series of measures to guide enterprises to save energy and reduce emissions through market mechanisms, such as environmental taxes and emissions trading. The voluntary participation type refers to the government's guidance or its own environmental awareness and voluntary environmental protection by enterprises and the public, thus forming constraints on enterprise pollution emissions, including voluntary environmental agreements, information disclosure, and public green consumption.

## 2.2. Research on the Quantification of Environmental Regulation

The measurement index systems of environmental regulation adopted in empirical studies have mainly comprised a single index system and a comprehensive index system. A single index system is composed of the same type of index. A comprehensive index system is composed of indicators from different dimensions or perspectives. A single index system can be roughly divided into three perspectives. One is the perspective of investment in pollution control. It was believed that the higher the investment in pollution control and emission reductions, the more the local government will pay attention to the governance and improvement of environmental problems [7,8]. Based on the Pollution Abatement Costs and Expenditures (PACE) survey, Levinson used pollution abatement expenditure to measure the intensity of environmental regulation in different industries in the United States [9]. Another perspective consists of the measurement of the intensity of environmental regulation from the angle of pollutant discharge or pollutant removal

rate. Scholars who have used this method believe that the high intensity of pollutant discharge usually represents the relatively loose intensity of environmental regulation [10,11]. Marconi studied the greenhouse gas (GHG) emission index produced by manufacturing and construction enterprises and reported that countries with lower GHG emission index values had stronger environmental regulation [12]. Korhonena et al. evaluated the effect of environmental regulations through reductions in sulfide emissions [13]. Other scholars used the ratio of pollutant emission to industrial added value to measure environmental regulation [14]. The third perspective is to measure the intensity of environmental regulations by the number of environmental laws and regulations issued by the government and the intensity of enforcement. Alpay et al. selected the number of environmental inspections of relevant departments reported by the media as a proxy indicator to measure the intensity of environmental regulation in Mexico [15]. Azzam et al. investigated the environmental legislation and standards of states in the United States to measure the intensity of environmental regulations, and they determined whether there were certain types of regulations in each year by reviewing the legislation of each state [16]. Ahmed et al. also measured environmental regulation in terms of patents related to environmental technologies [17]. The comprehensive index evaluated the intensity of environmental regulation by constructing a complex index system. Murty and Kumar constructed a proxy index to measure the intensity of environmental regulation with an RI (regulation index) and CI (water conservation index) [18]. Xing, X.P. et al. constructed a comprehensive index to measure the intensity of environmental regulation through four items: strict standards, national conditions, clarity, and effectiveness in solving environmental problems [19].

### 2.3. Spatial Correlation Analysis of Environmental Regulation

2.3.1. Spatial Correlation Theory

Spatial correlation refers to the correlation between something (e.g., economic factors, environment, and policy making) in a specific geographical unit and surrounding geographical units. Things that are close to each other generally tend to exhibit similar or identical characteristics. This kind of convergence in spatial distribution has been proposed by more and more scholars and attracted the attention of all sectors of society. According to the clustering method and analysis results, spatial correlation can be divided into positive and negative correlation. When a large number of attribute values of a certain thing gather together, presenting a convergent geographical distribution pattern, a positive spatial correlation is presented [20]. Economic and social factors in different regions are interrelated and influence each other [21]. The economic development level of adjacent areas is similar to a certain extent, which leads to a certain regional correlation between energy consumption and environmental pollution [22].

Otherwise, when the attribute values of a certain thing are widely dispersed, showing different geographical distribution patterns, a negative spatial correlation is presented. China is a vast country with many provinces. There are obvious differences in geographical characteristics, social and economic development levels, and urban spatial distribution among different regions. There are also differences in environmental pollution in different provinces and regions. The homogeneity and differences of the environmental pollution states between different regions lead to the correlations and "positive externalities" of environmental supervision policies in different regions. If a certain area implements a more stringent environmental control policy, it can not only effectively reduce local environmental pollution but also improve the environmental quality of neighboring areas.

Based on the relationship between whole and part, spatial correlation can be divided into global and local spatial correlation, respectively. Global spatial correlation means that a whole region has the characteristics of agglomeration on the whole. The correlation of local space means that the analysis objects in a particular region have a certain agglomeration with their adjacent research objects.

On the basis of studying the spatial correlation of various economic and social factors, the spatial econometric model has been widely used in economics, management, sociology

and other fields [23]. Compared to general econometrics, spatial econometrics has the advantage of introducing and considering spatial effects. In the past, with the help of spatial econometric models, scholars mainly studied how to explore the spatial correlation and spatial heterogeneity of research objects with regression analysis models of various data types. For example, Paelinck and other scholars pointed out in the book *Spatial Econometrics* that spatial econometrics mainly involves the problems of the spatial correlation, spatial difference, and spatial econometric model of research objects [24]. Anselin conducted a systematic study of the spatial econometric model on the basis of previous studies and constructed an initial overall research framework. He believed that the spatial econometric model is an econometric method that can be used to screen, calculate, verify and predict key influencing factors on the premise of acknowledging the existence of spatial effects of research objects [25].

2.3.2. Spatial Correlation Analysis of Environmental Regulation in China

The degrees of environmental pollution and economic development greatly vary in different regions of China, leading to great differences in the importance of environmental governance and environmental supervision among provinces. The efficiency of environmental regulation gradually increases with improvements of the level of economic development. The efficiency of the environmental regulation in the eastern region of China is higher than that in the western region [26]. After further subdividing the regions, it was found that the eastern region has the highest environmental regulation, the northeast region has the second highest environmental regulation, the central region has the third highest environmental regulation, and the western region has the lowest environmental regulation. The spatial agglomeration of environmental regulation efficiency is obvious [27].

Some scholars have divided China into different urban agglomerations and analyzed the spatial correlation of environmental regulations in different urban agglomerations. The Beijing–Tianjin–Hebei urban agglomeration pays attention to the adjustment of industrial structure, and the efficiency of environmental regulation has most significantly increased. The Yangtze River Delta urban agglomeration pays attention to terminal governance, and the efficiency of environmental regulation has shown a significant downward trend. The PEARL River Delta urban agglomeration pays attention to the introduction of cleaner production technology, and its efficiency of environmental regulation is clearly better than in the two other urban agglomerations [28].

*2.4. Factors Affecting the Spatial Relevance of Environmental Regulations*

First of all, the level of economic development is primary concern of many scholars. Orubu and Ben verified that there was an EKC curve relationship between economic growth and the ecological environment at different stages of economic development [29,30]. The growth of economic activities and resource demand could destroy the environment [31], but economic growth could also reduce pollution emissions through the application of more environmentally friendly technologies and the adjustment of industrial structures [32]. Therefore, the difference of economic development levels is an important factor affecting the intensity of environmental regulation in different regions. Secondly, industrial structure has been claimed as the key factor to solve economic development and environmental problems. Grossman and Krueger believed that structural effects could improve the environment [33], and reductions in the proportion of the output value of the secondary industry in GDP could significantly reduce $NO_2$ and $SO_2$ pollution [34]. Therefore, difference of industrial structures, especially the level of industrialization, will also significantly affect the intensity of environmental regulation in each region [35]. Thirdly, urbanization could also affect environmental policy. One view holds that the process of urbanization might cause negative environmental impacts, such as the heat island effect, the greenhouse effect, the deterioration of water quality, and other environmental pollution problems [36]. However, a different view holds that the process of urbanization could promote the improvement of labor productivity and economic structure. Thus, the process of urbanization could affect

the implementation effect of environmental regulations in various regions and alleviate the increasing trend of industrial pollutant emissions [37]. Fourthly, the sustainable development of the environment should be consistent with environmental education [38] because higher education could increase the incentive for environmental improvement and increase the government's attention to local environmental issues [39]. In addition, the scale of enterprises, residents' awareness of pollution control, and the level of foreign investment will affect the formulation and implementation of environmental regulations in various regions [40].

## 3. Research Design

### 3.1. Index Selection and Data Sources

The market alone cannot effectively solve the dilemma between environmental quality improvement and high-quality economic development because it cannot encourage enterprises to attach importance to environmental protection and engage in green production. Government intervention is needed to solve the problem of market failure. The local government's investment in local environmental governance reflects the government's emphasis on environmental improvement and the intensity of environmental regulation, as well as directly determining the improvement degree of local environmental quality. Therefore, referring to the research results of Song et al., the authors of this paper took the proportion of the total investment in local environmental pollution control in GDP as the measurement index of environmental regulation intensity (ERI) to reflect the expenditure on local pollution control [41]; the larger the index value is, the stronger the local environmental regulation.

Considering the availability of data, the authors of this paper used 30 provinces (autonomous regions and municipalities directly under the central government) in China, excluding Hong Kong SAR, Macao SAR, Taiwan Province, and Tibet Autonomous Region, as research samples to verify the spatial differences and correlations of environmental regulation intensity in different regions of China (the eastern region was set to include Beijing, Tianjin, Hebei, Shanghai, Jiangsu, Zhejiang, Fujian, Shandong, Guangdong, and Hainan; the central region was set to include Shanxi, Anhui, Jiangxi, Henan, Hubei, and Hunan; the western region was set to include Guangxi, Chongqing, Sichuan, Guizhou, Yunnan, Shaanxi, Gansu, Qinghai, Ningxia, Xinjiang, and Inner Mongolia; and the northeast region was set to include Liaoning, Jilin and Heilongjiang). The data were mainly derived from China Statistical Yearbook, China Environmental Statistical Yearbook and China Urban Statistical Yearbook from 2011 to 2020.

### 3.2. Methodology

This paper was mainly aimed to achieve the following: (1) Use the Theil index to explore whether there are spatial differences in environmental regulation intensity in different regions of China; (2) use Goble Moran's I index to explore whether there is a spatial correlation between the intensity of environmental regulations in different regions, i.e., whether the environmental regulation intensity of a province affects the formulation of environmental regulation policies in other neighboring areas, and whether the formulation of provincial environmental regulation policies is affected by the intensity of environmental regulation of neighboring enterprises; (3) if there is a spatial correlation between the intensity of environmental regulation in different regions, investigate the probability of each province changing the environmental regulation intensity in the process of transferring the environmental regulation intensity due to the influence of the environmental regulation intensity in neighboring provinces.

#### 3.2.1. The Theil Index

At present, there are many methods to measure regional differences, including the variation coefficient, difference coefficient, Gini coefficient, and the Theil index method. Different methods have different effects and scope of application.

According to the research of Theil, Slime and Hammami [42,43], the greatest advantage of using the Theil index method to measure spatial differences is that it can measure the contribution of intra-regional differences and inter-regional differences in reference to the overall differences. Compared to other methods, the Theil index method has comprehensive advantages. According to the geographical location of different provinces, the authors of this paper divided China into four regions, namely the eastern region, the central region, the western region, and the northeastern region. The Theil index was used to verify whether there were spatial differences in the environmental regulation intensity in these four regions and their differences in size. In order to analyze the spatial differences of environmental regulation intensity in China in detail, the Theil index was further divided into intra-regional differences and inter-regional differences. Intra-regional difference refers to whether there is a difference in the environmental regulation intensity in the provinces within each region and the size of the difference. Inter-regional difference refers to whether there is a difference in environmental regulation intensity between the four regions and the size of difference.

The Theil index is calculated as follows:

$$Theil = T_{inter} + T_{intra} \tag{1}$$

$$T_{inter} = \sum_{i=1}^{n_e} T_i ln(n_e \frac{T_i}{T_e}) + \sum_{i=1}^{n_c} T_i ln(n_c \frac{T_i}{T_c}) + \sum_{i=1}^{n_w} T_i ln(n_w \frac{T_i}{T_w}) + \sum_{i=1}^{n_{ne}} T_i ln(n_{en} \frac{T_i}{T_{ne}}) \tag{2}$$

$$\begin{aligned} T_{intra} = ln(T_e \frac{n}{n_e})\sum_{i=1}^{n_e} T_i ln(n_e \frac{T_i}{T_e}) + ln(T_c \frac{n}{n_c})\sum_{i=1}^{n_c} T_i ln(n_c \frac{T_i}{T_c}) \\ + ln(T_w \frac{n}{n_w})\sum_{i=1}^{n_w} T_i ln(n_w \frac{T_i}{T_w}) + ln(T_{ne} \frac{n}{n_{ne}})\sum_{i=1}^{n_{ne}} T_i ln(n_{ne} \frac{T_i}{T_{ne}}) \end{aligned} \tag{3}$$

The meanings of each symbol in Formulas (1)–(3) are listed in Table 2.

**Table 2.** Table of meanings of symbols in the Theil index calculation.

| Symbol | Variable | Name |
|---|---|---|
| $T_{inter}$ | Inter-regional difference | The spatial differences of environmental regulation intensity in the east, central, west and northeast regions |
| $T_{intra}$ | Intra-regional difference | The spatial differences of environmental regulation intensity among provinces in each region |
| $n$ | Number | The number of provinces; $n = 30$ |
| $n_e/n_c/n_w/n_{ne}$ | Number | The number of provinces in the eastern/central/western/northeastern regions, respectively. The values of $n_e/n_c/n_w/n_{ne}$ are 10/6/11/3, respectively. |
| $T_i$ | Weight | The proportion that the environmental regulation intensity of province $i$ is divided by the sum of the environmental regulation intensity of 30 provinces. |
| $T_e/T_c/T_w/T_{ne}$ | Weight | The proportion that the environmental regulation intensity of the eastern/central/western/northeastern region is divided by the sum of the environmental regulation intensity of 30 provinces. |

### 3.2.2. Spatial Autocorrelation Analysis

Spatial autocorrelation means that the attribute data of a geographical unit are consistent or opposite to the attribute data of their surrounding geographical units, that is, under the influence of spatial interaction and spatial diffusion, the attribute data of different geographical units are no longer independent but interact with each other. Through spatial autocorrelation analysis, the authors of this paper investigated whether the environmental regulation intensity in 30 provinces (autonomous regions and municipalities directly under the central government) in China was spatially dependent, that is, whether the intensity of environmental regulation in each province was internally correlated with the formulation of environmental regulation policies in other provinces, as well as the degree and direction of the correlation.

Common spatial autocorrelation analysis includes global and local autocorrelation. Global autocorrelation refers to the agglomeration and spatial correlation of a whole region. Local autocorrelation means that the analysis objects in a particular region have a certain agglomeration with the research objects in adjacent regions. At this point, it is not the whole region but some provinces in the region that have a spatial correlation. Because the environmental regulation intensity has the characteristics of space, practice, and diffusion and the administrative policies of neighboring provinces in China have homogeneity and promotion effects, environmental regulation policies have mutual influence among different regions. Therefore, the average correlation degree and significance of environmental regulation intensity among different geographical units can be reflected through global autocorrelation analysis, and the spatial distribution of environmental regulation intensity can be revealed as a whole.

Goble Moran's I index is commonly used to test global spatial autocorrelation. The specific calculation formula of Goble Moran's I index is as follows:

$$Moran's\ I = \frac{\sum_{i=1}^{n}\sum_{j=1}^{n} w_{ij}(x_i - \overline{x})(x_j - \overline{x})}{S^2 \sum_{i=1}^{n}\sum_{j=1}^{n} w_{ij}} \tag{4}$$

In Formula (4), $S^2 = \frac{1}{n}\sum_{i=1}^{n}(x_i - \overline{x})^2$, $\overline{x} = \frac{1}{n}\sum_{i=1}^{n} x_i$. The definitions of other symbols are shown in Table 3.

**Table 3.** Table of meanings of symbols in Goble Moran's I index calculation.

| Symbol | Variable | Name |
|---|---|---|
| $x_i$ | Environmental regulation intensity of province $i$ | The proportion of total investment in local environmental pollution control in GDP of province $i$ |
| $\overline{x}$ | The average value | The spatial differences of environmental regulation intensity in east, central, west and northeast regions |
| $n$ | Number | The number of provinces; $n = 30$ |
| $w_{ij}$ | Spatial weight matrix | The spatial weights of elements I and J |

In this paper, the Moran test was performed with the Stata software. The *p*-value and Z value were used to judge whether there was spatial correlation of environmental regulation intensity. The *p*-value was used to reflect the probability of a spatial autocorrelation of environmental regulation intensity. The Z value was used as the standard score. If the *p*-value was less than 10%, the Moran test passed the significance test. If so, the environmental regulation intensity of each province has a spatial autocorrelation, the environmental regulation intensity in a province will affect the formulation of environmental policies in neighboring provinces, and the change of environmental policies in neighboring provinces will also affect the environmental regulation intensity in their own province.

Moran's I index ranges from [−1, 1]. Here, a positive value of Moran's I index indicated that there was a positive spatial correlation among regions, that is, provinces with similar environmental regulation intensity had agglomeration. With a positive value, the change of environmental regulation intensity in one province is positively correlated with the change trend of environmental regulation in neighboring provinces. When the intensity of environmental regulation in a province increases, neighboring provinces will pay more attention to environmental governance and improve the intensity of environmental regulation. At the same time, strengthening environmental policy management in neighboring provinces will lead to an increase in environmental regulation intensity in the province; A negative value of Moran's I index indicates a negative spatial correlation, that is, provinces with similar environmental regulation intensity are dispersed. The change of environmental regulation intensity in a province is negatively correlated with the change trend of environmental regulation in neighboring provinces. When the intensity of environmental

regulation increases in one province, it will lead to the relaxation of environmental policy management in neighboring provinces. At the same time, strengthening environmental policy management in neighboring provinces will lead to the decline of environmental regulation intensity in the province; when Moran's I index equals 0, there is no spatial autocorrelation. With an index of 0, the environmental regulation intensity of a province does not affect the formulation of environmental policies in neighboring regions or has a low degree of influence, and is not affected by the intensity of environmental regulation in neighboring regions. The larger the absolute value of Moran's I index is, the stronger the spatial correlation. The variation of environmental regulation intensity in one province leads to a greater variation of environmental regulation in neighboring provinces.

### 3.2.3. Spatial Markov Chains

The Markov chain was proposed by Andrei Markov, a Russian mathematician, to reflect the development states and change trends of things by constructing a probability distribution matrix of the mutual transformation of different states for a group of discrete times and states. A traditional Markov chain can be used to investigate the probability of a province's environmental regulation transferring from one intensity to another intensity during a given period and to analyze the state of the transfer of provincial environmental regulation intensity, including whether the environmental regulation intensity of a province is stable and unchanged, whether the intensity is increased or decreased, and how long the transfer intensity span is. However, this process is affected by geographical unit location, as the environmental regulation intensity of a region is not independent, memorized, or completely random—rather, it is affected by the regional phenomena of adjacent geographical units. Therefore, the concept of spatial lag was introduced to investigate the transfer probability of the attribute data of a geographical unit under different geographical background conditions considering spatial proximity factors, i.e., the probability that the intensity of environmental regulation will increase or decrease after considering the influence of the environmental regulation intensity in neighboring provinces.

According to the quartiles, the environmental regulation intensity of each province could be divided into different levels. The spatial lag value of a province is represented by the domain state of the province, and the spatial weight matrix of each geographical unit is constructed. The M × M order state transition probability matrix of the traditional Markov chain was decomposed into M × M transition conditional probability matrices to analyze the internal relationship between the increase or decrease in environmental regulation intensity and the spatial geographical background so as to analyze the spatial spillover law of environmental regulation intensity in the regions and their internal provinces. In this paper, we classified the lag conditions according to the spatial lag efficiency of the environmental regulation intensity in 30 provinces (autonomous regions and municipalities directly under the central government) in the initial years. The spatial lag condition was calculated by the product of environmental regulation intensity and spatial weight matrix, namely $\sum W_{ij}X_j$, where $X_j$ represents the environmental regulation intensity of a certain region and $W_{ij}$ represents the element of spatial weight matrix, as shown in Table 4.

By comparing the traditional Markov transition probability matrix and the spatial Markov transition probability matrix, the internal relationship between the increase or decrease in environmental planning intensity and the spatial geographical background could be analyzed, and the spatial spillover law of environmental regulation intensity in the regions could be analyzed too.

**Table 4.** State transition matrix of spatial Markov chains (assuming M = 4).

| Spatial Lag | t/t + 1 | 1 | 2 | 3 | 4 |
|---|---|---|---|---|---|
| 1 | 1 | $X_{11/1}$ | $X_{12/1}$ | $X_{13/1}$ | $X_{14/1}$ |
| | 2 | $X_{21/1}$ | $X_{22/1}$ | $X_{23/1}$ | $X_{24/1}$ |
| | 3 | $X_{31/1}$ | $X_{32/1}$ | $X_{33/1}$ | $X_{34/1}$ |
| | 4 | $X_{41/1}$ | $X_{42/1}$ | $X_{43/1}$ | $X_{44/1}$ |
| 2 | 1 | $X_{11/2}$ | $X_{12/2}$ | $X_{13/2}$ | $X_{14/2}$ |
| | 2 | $X_{21/2}$ | $X_{22/2}$ | $X_{23/2}$ | $X_{24/2}$ |
| | 3 | $X_{31/2}$ | $X_{32/2}$ | $X_{33/2}$ | $X_{34/2}$ |
| | 4 | $X_{41/2}$ | $X_{42/2}$ | $X_{43/2}$ | $X_{44/2}$ |
| 3 | 1 | $X_{11/3}$ | $X_{12/3}$ | $X_{13/3}$ | $X_{14/3}$ |
| | 2 | $X_{21/3}$ | $X_{22/3}$ | $X_{23/3}$ | $X_{24/3}$ |
| | 3 | $X_{31/3}$ | $X_{32/3}$ | $X_{33/3}$ | $X_{34/3}$ |
| | 4 | $X_{41/3}$ | $X_{42/3}$ | $X_{43/3}$ | $X_{44/3}$ |
| 4 | 1 | $X_{11/4}$ | $X_{12/4}$ | $X_{13/4}$ | $X_{14/4}$ |
| | 2 | $X_{21/4}$ | $X_{22/4}$ | $X_{23/4}$ | $X_{24/4}$ |
| | 3 | $X_{31/4}$ | $X_{32/4}$ | $X_{33/4}$ | $X_{34/4}$ |
| | 4 | $X_{41/4}$ | $X_{42/4}$ | $X_{43/4}$ | $X_{44/4}$ |

## 4. Spatial Difference Analysis of Environmental Regulation Intensity in China

In this paper, we discuss the spatial differences of environmental regulation intensity in China from three perspectives: the temporal variation trend of environmental regulation intensity in China, whether there are spatial differences among the four regions, and whether there are differences within regions.

### 4.1. Temporal Variation Trend Analysis of Environmental Regulation Intensity in China

The intensity of environmental regulation was measured by the proportion of total investment in environmental pollution control in GDP of each province. The mean value of environmental regulation intensity in 30 provinces was used to measure the intensity of environmental regulation in China, and the changing trend of environmental regulation intensity in China was judged. Figure 1 reflects the changing trend of environmental regulation intensity in China from 2010 to 2019. According to Figure 2, from 2010 to 2013, the intensity of environmental regulation in China increased from 1.52% to 1.79% of the total GDP. After 2013, the intensity of China's environmental regulations showed a downward trend. The share of total investment in environmental pollution control in China's GDP dropped from 1.79% in 2013 to 1.18% in 2019, a decline rate of 34.08%.

(1) From 2010 to 2013, the intensity of China's environmental regulation increased from 1.52% in 2010 to 1.79% in 2013, indicating that the intensity of China's environmental regulation increased and provinces paid more attention to the improvement of environmental quality. This intensity peaked in 2013 with a 17.76% increase. The reason for this increase may be that in the 21st century, China's economic development level has been greatly improved and the policy concept, policy objectives and policy means of environmental regulation have been constantly changing. The Chinese government has issued a large number of policies related to environmental protection. Local governments are also paying more attention to environmental quality and have issued a large number of local laws and regulations to promote environmental quality improvement. Total investment in environmental pollution control increased from 665.42 billion yuan in 2010 to 951.65 billion yuan in 2013, and the intensity of environmental regulations also increased.

(2)  From 2014 to 2019, the intensity of environmental regulation decreased compared to all previous years except for 2015. The main reason for the decrease in the intensity of environmental regulation was the overall reduction in the emission of major pollutants in China, as the emission of industrial waste water decreased by 9.15% and the emission of industrial $SO_2$ decreased by 64.97%, indicating that China's environmental governance achieved remarkable results. The ecological environment quality has continuously improved, and environmental quality has basically reached the standard. The improvement of environmental quality has led to the relaxation of environmental governance supervision by local governments and the decline of environmental regulation intensity. The increase in environmental regulations in 2016 occurred because China's Environmental Protection Law, which came into effect in 2015, further strengthened the punishment of local governments for local environmental violations, leading to an increase in the intensity of environmental regulations nationwide in 2015.

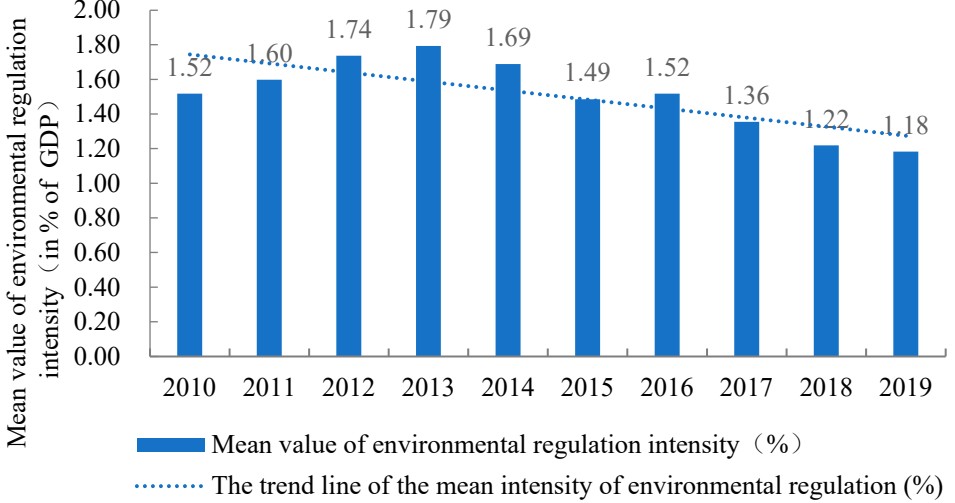

**Figure 2.** The changing trend of the environmental regulation intensity in China from 2010 to 2019.

*4.2. Spatial Difference Analysis of Environmental Regulation Intensity in China*

In order to better analyze the spatial differences of environmental regulation intensity in China, we divided China into the eastern, central, western, and northeastern regions according to the traditional criteria of four economic regions in China. The Theil index of the country and the four major economic regions was further calculated to measure the overall differences of China's environmental regulation intensity, the inter-regional differences (i.e., the differences of environmental regulation intensity among four regions), and the intra-regional differences (i.e., the differences of environmental regulation intensity among provinces within the regions). Then, we identified whether the overall environmental regulation differences in China were caused by the intra-regional or inter-regional differences, as shown in Table 5 and Figure 3.

**Table 5.** The Theil index of regional environmental regulation intensity.

| Year | Intra-Regional Differences | Proportion (%) | Inter-Regional Differences | Proportion (%) | Overall Differences |
|------|---------|---------|---------|---------|---------|
| 2010 | 0.1044 | 89.70 | 0.0120 | 10.30 | 0.1163 |
| 2011 | 0.1213 | 90.16 | 0.0132 | 9.84 | 0.1346 |
| 2012 | 0.1205 | 79.54 | 0.0310 | 20.46 | 0.1515 |
| 2013 | 0.1029 | 88.89 | 0.0129 | 11.11 | 0.1157 |
| 2014 | 0.1208 | 89.35 | 0.0144 | 10.65 | 0.1352 |
| 2015 | 0.1041 | 86.68 | 0.0160 | 13.32 | 0.1201 |
| 2016 | 0.1370 | 89.38 | 0.0163 | 10.62 | 0.1533 |
| 2017 | 0.1162 | 88.51 | 0.0151 | 11.49 | 0.1313 |
| 2018 | 0.1959 | 91.04 | 0.0193 | 8.96 | 0.2152 |
| 2019 | 0.0980 | 87.88 | 0.0135 | 12.12 | 0.1116 |

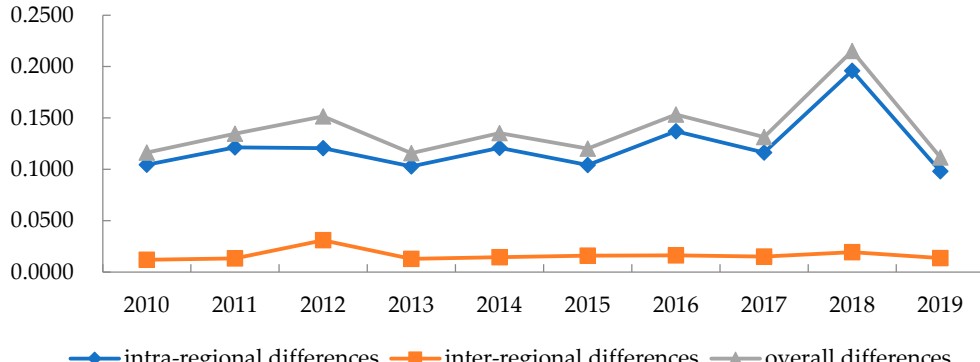

**Figure 3.** The Theil index of regional environmental regulation intensity.

In Table 5, the values of the intra-regional differences in column 2 were calculated according to Equation (3). In Equation (3), $T_{intra}$ represents the intra-regional differences. The values of Proportion in column 3 were obtained by the proportion of $T_{intra}$ to the Theil index. According to Equation (1), the value of the Theil index is the sum of $T_{intra}$ and $T_{inter}$. The values of the inter-regional differences in column 4 were calculated according to Equation (2). In Equation (2), $T_{inter}$ represents the inter-regional differences. The values of Proportion in column 5 were obtained by the proportion of $T_{inter}$ to the Theil index. Therefore, the sum of the values in columns 3 and 5 is equal to 100%. The values of the overall differences in column 6 are the Theil index for that year. According to Equation (1), the value of the Theil index is the sum of $T_{intra}$ and $T_{inter}$. As shown in Table 5 and Figure 2, intra-interval differences accounted for more than 85% of the overall differences, indicating that intra-interval difference was the main reason for the spatial difference of environmental regulation intensity in China.

*4.3. Inter-Regional Differences of Environmental Regulation Intensity in China*

In order to further analyze the reasons for the great fluctuation of the spatial difference of environmental regulation intensity in China, the authors of this paper successively measured the environmental regulation intensity of the four major economic regions according to the mean value of the environmental regulation intensity of all provinces in each economic region, as shown in Table 6 and Figure 4. It was found that there were significant differences in the environmental regulation intensity in the four regions. On the whole, a decreasing trend of "west–central–northeast–east" was observed.

**Table 6.** Variation trend of environmental regulation intensity in four regions (%).

| Regions | 2010 | 2011 | 2012 | 2013 | 2014 | 2015 | 2016 | 2017 | 2018 | 2019 | Mean |
|---|---|---|---|---|---|---|---|---|---|---|---|
| The eastern region | 1.49 | 1.34 | 1.35 | 1.38 | 1.41 | 1.06 | 1.06 | 1.07 | 0.86 | 1.22 | 1.22 |
| The central region | 1.25 | 1.43 | 1.66 | 1.64 | 1.43 | 1.49 | 1.82 | 1.41 | 1.12 | 1.29 | 1.45 |
| The western region | 1.65 | 1.89 | 1.97 | 2.25 | 2.19 | 1.92 | 1.90 | 1.69 | 1.72 | 1.24 | 1.84 |
| The northeastern region | 1.37 | 1.39 | 1.50 | 1.51 | 1.42 | 1.27 | 1.44 | 1.24 | 0.99 | 1.25 | 1.34 |
| The national | 1.44 | 1.51 | 1.62 | 1.69 | 1.61 | 1.44 | 1.55 | 1.35 | 1.17 | 1.25 | 1.46 |
| The standard deviation | 0.170 | 0.255 | 0.266 | 0.383 | 0.382 | 0.369 | 0.387 | 0.262 | 0.381 | 0.028 | 0.170 |

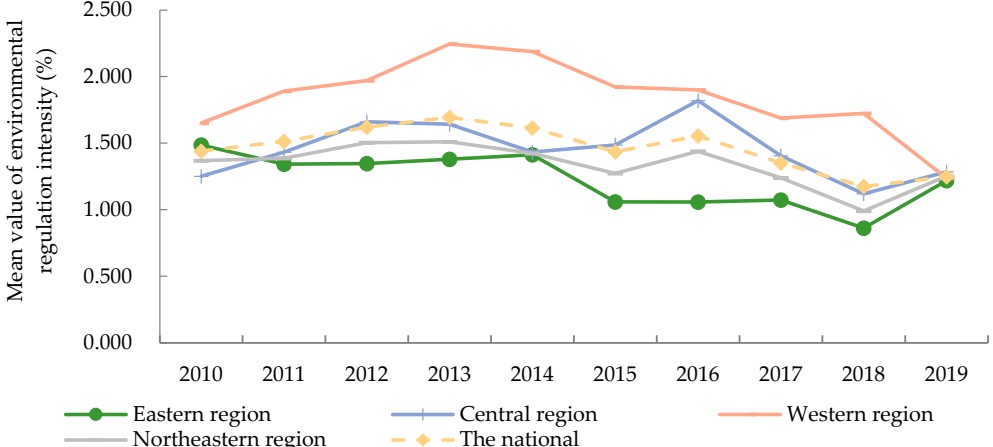

**Figure 4.** Variation trend figure of environmental regulation intensity in four regions.

4.3.1. Analysis of the Changing Trend of the Environmental Regulation Intensity in the Western Region

The environmental regulation intensity in the western region was found to be much higher than the national average and higher than in other regions in the same period except for 2012, when it was lower than that in the northeastern region. The main reason for this result was that the western region is mostly in the early and middle stages of industrialization and its degree of industrialization is low. Industrial structure is an important factor that affect the intensity of environmental regulation. As the manufacturing process is prone to produce a large number of pollutants, a large proportion of manufacturing in GDP will produce more pollution emissions, posing a threat to local environmental governance [44]. Due to the strong demand of economic development, the environmental regulation intensity in the early stage was at a low level, which led to the increasingly prominent environmental problems. In recent years, with the implementation of the "Western Development" strategy and the "One Belt and One Road" initiative, provincial governments in the western region have increased the investment of environmental pollution control capital and improved the intensity of environmental regulation to balance the relationship between economic development and ecological environment.

4.3.2. Analysis of the Changing Trend of the Environmental Regulation Intensity in the Central Region

The environmental regulation intensity in the central region was found to be closest to the national average. It was higher than the national average in 2016, mainly because in 2016, the National Development and Reform Commission designated the six central provinces as national demonstration areas for ecological civilization construction in The 13th Five-Year Plan of Promoting the Rise of The Central Region. In order to better play the pivotal role of coordinated development between the east and the west and to make a good link between industrial transfer channels, local governments have raised the cost of environmental standards.

### 4.3.3. Analysis of the Changing Trend of the Environmental Regulation Intensity in the Eastern Region

The intensity of environmental regulation in the eastern region was found to be lower than that in other regions. The main reasons are as follows. First, the eastern region has superior geographical environment and economic development levels. Resource demands accompanying economic growth can destroy the environment, and the technological innovation promoted by economic growth can promote pollution control and emission reductions [45]. With a long history of resource development and utilization, the local government has paid attention to environmental problems from an earlier point and has a strong awareness of fulfilling environmental responsibility and outstanding environmental improvement effects. Therefore, the high speed and level of economic development will promote the high intensity of environmental regulation in the eastern region in the future. Second, the eastern region is densely populated and has a high level of urbanization. On the one hand, the population agglomeration brought by the improvement of urbanization level will increase resource exploitation and environmental pollution; on the other hand, it will drive industrial transformation and promote the continuous improvement of environment through talent introduction and labor force improvement [46]. Third, with the in-depth development of industrial transfer and industrial structure transformation in the eastern region, pollution-intensive industries in the eastern region have been gradually transferred to the central region and the western region, resulting in the annual reduction in pollution emissions in the eastern region and effective alleviation of environmental pressure.

### 4.3.4. Analysis of the Changing Trend of the Environmental Regulation Intensity in the Northeast Region

The intensity of environmental regulation was found to decrease the most in the northeast region; it was higher than the national average from 2010 to 2012 but has been at the lowest levels since 2016. There are two reasons for the decline of environmental regulation intensity in the northeast region. First, the northeast region is the heavy industry base of China. Heavy industries such as oil exploitation and mineral exploitation with high energy consumption are important pillar industries in the northeast region. This industrial structure gradually destroyed the local natural resources and ecological environment. Under the background at that time, local government gradually improved the environmental regulation system and reformed in key industries for energy conservation and emissions reduction. However, the vitality of economic development was therefore restricted. In 2015, the government strategically decided to comprehensively revitalize the old industrial base in the northeast region. The local government reduced the environmental regulation intensity to reduce the production cost of enterprises. Second, the northeast has recently suffered a serious population loss, with a large number of high-end talents leaving the region, leading to a decline in the educational level of the region. Lower levels of education can reduce environmental awareness and further constrain governments' incentives to focus on local environmental problems. As a result, the intensity of environmental regulation implemented by provincial governments in northeast China has declined [47].

### 4.4. Analysis of the Intra-Regional Differences in China's Environmental Regulation Intensity

Based on the proportion of total investment in environmental pollution control in GDP of each province, the variation trend of environmental regulation intensity of 30 provinces (autonomous regions and municipalities directly under the central government) in China from 2010 to 2019 was obtained, as shown in Table 7.

**Table 7.** Environmental regulation intensity in % of total GDP in 30 provinces of China from 2010 to 2019.

| Region | Province | 2010 | 2011 | 2012 | 2013 | 2014 | 2015 | 2016 | 2017 | 2018 | 2019 |
|---|---|---|---|---|---|---|---|---|---|---|---|
| | Beijing | 0.0155 | 0.0124 | 0.0180 | 0.0205 | 0.0272 | 0.0166 | 0.0249 | 0.0223 | 0.0192 | 0.0164 |
| | Tianjin | 0.0161 | 0.0216 | 0.0174 | 0.0192 | 0.0262 | 0.0116 | 0.0047 | 0.0057 | 0.0031 | 0.0074 |
| | Hebei | 0.0206 | 0.0292 | 0.0211 | 0.0202 | 0.0181 | 0.0151 | 0.0140 | 0.0198 | 0.0145 | 0.0138 |
| | Shanghai | 0.0075 | 0.0072 | 0.0063 | 0.0081 | 0.0099 | 0.0082 | 0.0069 | 0.0049 | 0.0029 | 0.0046 |
| The Eastern | Jiangsu | 0.0113 | 0.0118 | 0.0122 | 0.0148 | 0.0136 | 0.0134 | 0.0099 | 0.0083 | 0.0075 | 0.0069 |
| Region | Zhejiang | 0.0122 | 0.0075 | 0.0109 | 0.0105 | 0.0118 | 0.0101 | 0.0138 | 0.0086 | 0.0076 | 0.0066 |
| | Fujian | 0.0086 | 0.0111 | 0.0110 | 0.0126 | 0.0078 | 0.0086 | 0.0064 | 0.0066 | 0.0078 | 0.0077 |
| | Shandong | 0.0143 | 0.0157 | 0.0172 | 0.0179 | 0.0162 | 0.0125 | 0.0133 | 0.0151 | 0.0137 | 0.0099 |
| | Guangdong | 0.0308 | 0.0063 | 0.0046 | 0.0056 | 0.0044 | 0.0039 | 0.0045 | 0.0040 | 0.0030 | 0.0047 |
| | Hainan | 0.0117 | 0.0114 | 0.0160 | 0.0085 | 0.0061 | 0.0059 | 0.0074 | 0.0120 | 0.0069 | 0.0439 |
| | Shanxi | 0.0232 | 0.0228 | 0.0281 | 0.0281 | 0.0242 | 0.0218 | 0.0440 | 0.0192 | 0.0145 | 0.0223 |
| | Anhui | 0.0136 | 0.0164 | 0.0180 | 0.0246 | 0.0190 | 0.0185 | 0.0189 | 0.0170 | 0.0115 | 0.0135 |
| The Central | Jiangxi | 0.0167 | 0.0208 | 0.0247 | 0.0168 | 0.0148 | 0.0140 | 0.0170 | 0.0156 | 0.0158 | 0.0177 |
| Region | Henan | 0.0058 | 0.0062 | 0.0072 | 0.0091 | 0.0085 | 0.0080 | 0.0089 | 0.0143 | 0.0115 | 0.0106 |
| | Hubei | 0.0090 | 0.0130 | 0.0126 | 0.0100 | 0.0112 | 0.0081 | 0.0139 | 0.0117 | 0.0091 | 0.0088 |
| | Hunan | 0.0068 | 0.0067 | 0.0090 | 0.0099 | 0.0083 | 0.0188 | 0.0065 | 0.0065 | 0.0048 | 0.0042 |
| | Inner Mongolia | 0.0291 | 0.0419 | 0.0425 | 0.0445 | 0.0462 | 0.0414 | 0.0331 | 0.0282 | 0.0143 | 0.0164 |
| | Guangxi | 0.0192 | 0.0157 | 0.0169 | 0.0175 | 0.0147 | 0.0177 | 0.0127 | 0.0103 | 0.0088 | 0.0101 |
| | Chongqing | 0.0219 | 0.0255 | 0.0161 | 0.0133 | 0.0115 | 0.0087 | 0.0080 | 0.0111 | 0.0080 | 0.0085 |
| | Sichuan | 0.0052 | 0.0067 | 0.0075 | 0.0088 | 0.0100 | 0.0071 | 0.0088 | 0.0081 | 0.0085 | 0.0079 |
| The Western | Guizhou | 0.0066 | 0.0116 | 0.0102 | 0.0138 | 0.0186 | 0.0130 | 0.0100 | 0.0159 | 0.0114 | 0.0156 |
| Region | Yunnan | 0.0137 | 0.0125 | 0.0119 | 0.0154 | 0.0108 | 0.0094 | 0.0089 | 0.0077 | 0.0084 | 0.0072 |
| | Shaanxi | 0.0182 | 0.0126 | 0.0128 | 0.0139 | 0.0164 | 0.0134 | 0.0167 | 0.0146 | 0.0079 | 0.0098 |
| | Gansu | 0.0162 | 0.0124 | 0.0225 | 0.0293 | 0.0220 | 0.0187 | 0.0170 | 0.0122 | 0.0912 | 0.0163 |
| | Qinghai | 0.0149 | 0.0191 | 0.0158 | 0.0214 | 0.0162 | 0.0174 | 0.0249 | 0.0167 | 0.0063 | 0.0091 |
| | Ningxia | 0.0220 | 0.0297 | 0.0261 | 0.0311 | 0.0318 | 0.0337 | 0.0364 | 0.0264 | 0.0114 | 0.0212 |
| | Xinjiang | 0.0146 | 0.0203 | 0.0344 | 0.0380 | 0.0424 | 0.0310 | 0.0325 | 0.0345 | 0.0134 | 0.0140 |
| The | Liaoning | 0.0149 | 0.0230 | 0.0383 | 0.0181 | 0.0136 | 0.0144 | 0.0086 | 0.0101 | 0.0070 | 0.0055 |
| North-eastern | Jilin | 0.0194 | 0.0131 | 0.0119 | 0.0112 | 0.0098 | 0.0111 | 0.0081 | 0.0084 | 0.0072 | 0.0068 |
| Region | Heilongjiang | 0.0158 | 0.0154 | 0.0198 | 0.0252 | 0.0150 | 0.0134 | 0.0146 | 0.0107 | 0.0084 | 0.0075 |
| Mean value | | 0.0152 | 0.0160 | 0.0174 | 0.0179 | 0.0169 | 0.0149 | 0.0152 | 0.0136 | 0.0122 | 0.0118 |

4.4.1. Analysis of the Intra-Regional Differences in the Environmental Regulation Intensity in the Eastern Region

The spatial difference of environmental regulation intensity among provinces in the eastern region was found to have been gradually widening. The intensity of environmental regulation in Shanghai and Guangdong was relatively low; it gradually decreased from 2010 to 2014 and basically remained stable after 2015. The main reason is that Shanghai and Guangdong are the two provinces with the fastest economic development and the highest GDP growth rate in China, resulting in a relatively low proportion of the input cost of environmental governance in GDP. Hebei, Beijing, and Shandong were found to be the three provinces with the highest environmental regulation intensity in the eastern region. The intensity of environmental regulation was also found to be different from that of Shanghai and Guangdong. The main reason is that Beijing, Hebei, and Shandong are the provinces with the highest environmental pollution degrees in the eastern region, and the three provincial government departments have placed great pressure on environmental control and a high intensity of environmental regulation.

4.4.2. Analysis of the Intra-Regional Differences in the Environmental Regulation Intensity in the Central Region

The spatial difference of environmental regulation intensity among provinces in the central region was found to remain stable. The intensity of environmental regulation in most provinces was found to slightly fluctuate. The province with the highest intensity of environmental regulation was Shanxi. The main reason is that Shanxi is the largest coal-producing area in China and there are a large number of coal-development enterprises, which has led to a high degree of environmental pollution. The government has placed great

pressure on environmental governance and a high intensity of environmental regulation. The province with the biggest change of environmental regulation intensity was found to be Henan. The intensity of environmental regulation increased from 0.62% in 2010 to 1.06% in 2019, so the intensity of environmental regulation nearly tripled. This was due to the rapid economic development in Henan in recent years, which has led to more attention being paid to environmental protection.

4.4.3. Analysis of the Intra-Regional Differences in the Environmental Regulation Intensity in the Western Region

The degree of spatial difference of environmental regulation intensity in the western region remained stable from 2010 to 2017 but increased sharply in 2018. The main reason is that the environmental regulation intensity in Gansu significantly increased in 2018, when it was as high as 9.12%. As a result, the intensity of environmental regulation in different provinces in the western region showed great differences and the degree of spatial difference increased. In 2018, Gansu's environmental management expenditure significantly increased in two aspects: first, the number and growth of environmental protection investment in the completion and acceptance of construction projects in Gansu were the highest in China; second, the major projects in Gansu in 2018 were arranged to highlight poverty alleviation, ecological environmental protection, and scientific and technological innovation. The reasons for the significant increase in the intensity of environmental regulation in Gansu are as follows. First, the intensity of poverty alleviation relocation in inhospitable areas and undertaking industrial transfer in the east has been increasing, as has the demand for resources, agricultural and forestry land or urban construction land. Secondly, Gansu's ecological environment itself is relatively weak, so the government needs to pay more attention to ecological environment protection. Third, due to its special geographical location, Gansu has the political responsibility to build and maintain the ecological security barrier in the western region, and it needs to strengthen environmental governance. Fourth, the ecological environment damage in Qilian Mountain in Gansu was notified by the central government, and the Ministry of Environmental Protection required Gansu government to conduct all-out efforts to rectify the ecological environment damage and environmental pollution in 2018. Accordingly, there was a significant increase in the intensity of environmental regulations in Gansu in 2018.

4.4.4. Analysis of the Intra-Regional Differences in the Environmental Regulation Intensity in the Northeast Region

The spatial difference of environmental regulation intensity among provinces in the northeast region was found to be the smallest, and the difference gradually decreased. There was a small difference in the intensity of environmental regulation in the three provinces in the northeast region, mainly because the economic development degrees and industrial structure of the three provinces are basically the same. Additionally, the three provinces in the northeast region have experienced a large outflow of personnel, especially high-end talents, and a decline in urbanization and education level. Therefore, the three provinces in the northeast region were found to have similar economic and social status, leading to the consistency of all kinds of administrative policies and administrative means formulated by the three provincial governments, as well as a small spatial difference of environmental regulation intensity.

## 5. Analysis of Spatial Correlation and Spatial Transition Evolution

*5.1. Spatial Correlation Test of Environmental Regulation Intensity in China*

The analysis presented in this section illustrates the obvious spatial differences in the environmental regulation intensity in China. In particular, we found great intra-regional differences in the intensity of environmental regulation among provinces in the four regions. This result led us to a further question: are there spatial dependence and correlation between neighboring provinces with large differences in environmental regulation intensity? That is, will the intensity of environmental regulations in one province affect the formulation of

environmental regulations in neighboring provinces, and will this intensity be affected by the intensity of environmental regulations in neighboring provinces?

The authors of this paper used Global Moran's I index to investigate whether the intensity of environmental regulation in 30 provinces (autonomous regions and municipalities directly under the central government) in China was spatially dependent, that is, whether the intensity of environmental regulation in each province was internally correlated with the spatial distribution and the degree and direction of the correlation. According to whether the boundaries between provinces were adjacent to each other, a spatial weight matrix was constructed (if the boundaries between provinces were shown to be adjacent, the value was 1; otherwise, the value was 0) and standardized. On this basis, the Global Moran's I value of each province in different years was measured through the environmental regulation intensity data of the provinces. The measurement results are shown in Table 8.

**Table 8.** Moran's I value of environmental regulation intensity in China from 2010 to 2019.

| Indicator | 2010 | 2011 | 2012 | 2013 | 2014 | 2015 | 2016 | 2017 | 2018 | 2019 |
|---|---|---|---|---|---|---|---|---|---|---|
| Moran's I | 0.040 | 0.207 | 0.269 | 0.390 | 0.291 | 0.266 | 0.222 | 0.149 | −0.023 | −0.103 |
| Z | 0.613 | 0.039 | 2.528 | 3.543 | 2.770 | 2.592 | 2.142 | 1.528 | 0.238 | −0.669 |
| P | 0.270 | 0.021 | 0.006 | 0.000 | 0.003 | 0.005 | 0.016 | 0.063 | 0.406 | 0.252 |

As shown in Table 8, the *p*-value of Global Moran's I index of environmental regulation intensity in 2010, 2018 and 2019 was greater than 0.1, indicating that the spatial correlation of provincial environmental regulation intensity in these three years was not obvious. The Global Moran's I index values of the environmental regulation intensity of provinces in 2011, 2012, 2013, 2014, 2015, 2016, and 2017 were significant at the levels of 5%, 1%, 1%, 1%, 1%, 5%, and 10%, respectively. This shows that there was a significant spatial correlation between the environmental regulation intensity of provinces in these 7 years. It can therefore be stated that the environmental regulation intensity of provinces will significantly affect the environmental policy making of neighboring provinces, and the environmental policy making of neighboring provinces will also be affected by the intensity of provincial environmental regulation.

In addition, in the year with spatial correlation of environmental regulation intensity (i.e., the *p*-value was less than 0.1), the Global Moran's I index of China's environmental regulation intensity was positive, indicating that the intensity of environmental regulation in each province has a positive spatial correlation and a spatial aggregation effect. An increase in environmental regulation intensity in one province will lead to an increase in environmental regulation intensity in neighboring provinces. Similarly, an increase in environmental regulation intensity in neighboring provinces will also lead to stricter environmental policies in this province.

*5.2. Evolution Characteristics of Environmental Regulation Intensity Shift in China*

5.2.1. Traditional Markov Chain Test

The authors of this paper first investigated the probability matrix of China's environmental regulation intensity transition based on the traditional Markov chain. According to the quartiles, the intensity of environmental regulation in each province was divided into four categories: low level, medium-low level, medium-high level, and high level, which are represented by H = 1, 2, 3, and 4, respectively. The higher H is, the higher the intensity of environmental regulation. The intensity of environmental regulation was divided into four adjacent but not overlapping complete intervals: (0.0029, 0.0086], (0.0086, 0.0133], (0.0133, 0.0180], (0.0180, 0.0912]. The first-order traditional Markov transition probability matrix obtained by these four grades is shown in Table 9. The value on the main diagonal is the probability that each level of environmental regulation intensity maintains its own

level, and the value on the non-main diagonal of the matrix is the probability that different levels of environmental regulation intensity convert to each other.

**Table 9.** Traditional Markov transition probability matrix of environmental regulation intensity in China.

| t/t + 1 | n | 1 | 2 | 3 | 4 |
|---------|----|-------|-------|-------|-------|
| 1 | 62 | 0.742 | 0.210 | 0.016 | 0.032 |
| 2 | 69 | 0.275 | 0.507 | 0.174 | 0.043 |
| 3 | 67 | 0.045 | 0.254 | 0.478 | 0.224 |
| 4 | 72 | 0.028 | 0.069 | 0.264 | 0.639 |

Table 9 shows the following. (1) The intensity of provincial environmental regulation has the characteristics of maintaining the stability of the original state grade. Regarding the diagonal elements of the transition probability matrix, the probability values on the diagonal were found to be larger than the probability values on the non-diagonal, with a minimum value of 0.478 and a maximum value of 0.742, indicating that the intensity of environmental regulation in each province is at least 47.8% likely to remain unchanged at the original state level. In addition, the maximum probability on the non-diagonal line is 27.5%, indicating that the probability of provincial environmental regulation intensity shifting is not large. (2) The probability of provincial environmental regulation intensity transferring to adjacent levels is greater than the probability of cross-level transfer. The probability that the intensity of environmental regulation in grade 1 will be transferred to grade 2 was found to be 21%, which was higher than the probability that the intensity of environmental regulation will be transferred to grade 3 by 1.6%. There was a 17.4% chance that the level 2 environmental regulation intensity would be transferred to level 3, higher than the 4.3% chance that it would be transferred to level 4. This shows that the strength of environmental regulations in each region must consider the inherent limitations of local geographical environment, resource endowment, economic development level and other factors to avoid a large fluctuation. (3) Regarding the direction of transfer, the probability of downward transfer of environmental regulation intensity was found to be greater than that of upward transfer. The probability of provincial environmental regulation intensity transferring from the second level to the first level was found to be 27.5%, which was higher than the probability of transferring to the third level of 17.4%. The probability of provincial environmental regulation intensity transferring from the third level to the second level was found to be 25.4%, which was higher than the probability of transferring to the fourth level of 22.4%. This shows that in recent years, China's ecological civilization construction has achieved great success, as has the transformation and upgrading of industrial structure. In addition, the high-quality development of economy urgently needs to reach a new breakthrough point, thus resulting in a downward trend of environmental regulation intensity.

5.2.2. Spatial Markov Chain Test

Through spatial correlation analysis, the evolution of environmental regulation intensity of each province was found to be affected not only by its own internal factors but also by the intensity of environmental regulation of neighboring provinces. Therefore, spatial geographical factors were introduced into the traditional Markov transfer probability matrix to investigate the spatial Markov transfer probability under the influence of adjacent geographical background factors, further exploring the evolution law of environmental regulation intensity in various provinces in China. The results are shown in Table 10.

**Table 10.** Spatial Markov transition probability matrix of environmental regulation intensity in China.

| Spatial Lag | t/t + 1 | n | 1 | 2 | 3 | 4 |
|---|---|---|---|---|---|---|
| 1 | 1 | 24 | 0.833 | 0.125 | 0.000 | 0.042 |
|  | 2 | 13 | 0.385 | 0.308 | 0.308 | 0.000 |
|  | 3 | 12 | 0.083 | 0.250 | 0.667 | 0.000 |
|  | 4 | 10 | 0.100 | 0.100 | 0.400 | 0.400 |
| 2 | 1 | 15 | 0.867 | 0.067 | 0.000 | 0.067 |
|  | 2 | 23 | 0.304 | 0.478 | 0.217 | 0.000 |
|  | 3 | 15 | 0.000 | 0.400 | 0.400 | 0.200 |
|  | 4 | 6 | 0.167 | 0.000 | 0.333 | 0.500 |
| 3 | 1 | 18 | 0.556 | 0.389 | 0.056 | 0.000 |
|  | 2 | 19 | 0.316 | 0.684 | 0.000 | 0.000 |
|  | 3 | 22 | 0.091 | 0.227 | 0.364 | 0.318 |
|  | 4 | 17 | 0.000 | 0.118 | 0.294 | 0.588 |
| 4 | 1 | 5 | 0.600 | 0.400 | 0.000 | 0.000 |
|  | 2 | 14 | 0.071 | 0.500 | 0.214 | 0.214 |
|  | 3 | 18 | 0.000 | 0.167 | 0.556 | 0.278 |
|  | 4 | 39 | 0.000 | 0.051 | 0.205 | 0.744 |

Comparing Tables 9 and 10, it can be seen that: (1) The transfer of provincial environmental regulation intensity does not exist in isolation in space but will be affected by the intensity of environmental regulation in neighboring provinces, such as P12 = 21%, P12|1 = 12.5%, P12|2 = 6.7%, and P12 ≠ P12|1 ≠ P12|2. It can also be seen that the intensity of environmental regulation in neighboring provinces plays an important role in the evolution of regional environmental regulation intensity. That is, the spatial correlation has a significant impact on the evolution trend of environmental regulation intensity. (2) Under the geographical background of different levels of environmental regulation intensity, the spillover effects of environmental regulation intensity transfer in neighboring provinces are different. Under the influence of the lower level of environmental regulation intensity in neighboring provinces, the probability of the transfer of environmental regulation intensity to a higher level increases, such as P12|1 = 12.5%, P12|2 = 6.7%, and P12|1 > P12|2. On the contrary, under the influence of higher level environmental regulation intensity in neighboring provinces, the probability of the transfer of environmental regulation intensity to a higher level decreases, such as P23|2 = 21.7%, P23|3 = 0, and P23|3 < P23|2. If a province has strict supervision on local environmental issues and large investment in governance, the local environmental quality will be higher and neighboring provinces will enjoy the environmental dividend brought by the province, thus weakening the intensity of environmental regulation of neighboring provinces. In contrast, if a province has serious environmental problems and does not pay attention to local environmental governance, the environmental pollution in this region will also affect the environmental quality of neighboring provinces, leading to the improvement of environmental regulation in surrounding provinces.

As shown in Table 10, the four transition probability matrices under the four spatial lag conditions are different. This shows that the probability of local environmental regulation intensity being affected is also different when the intensity of environmental regulation in neighboring provinces is different. The intensity of environmental regulation in neighboring provinces plays an important role in the evolution of local environmental regulation intensity. That is, spatial correlation affects the evolution trend of environmental regulation intensity.

(1)     When the intensity of environmental regulation in neighboring provinces was at a low level, the local province was shown to have an at least 30.8% probability of maintaining the intensity of environmental regulation. When the intensity of environmental regulation of neighboring provinces was at a medium-low level, the local province was found to have an at least 40% probability of maintaining the intensity of environmental regulation. When the intensity of environmental regulation in neighboring provinces was at a medium-high level, local province was found to have an at least 36.4% probability of maintaining the intensity of environmental regulation. When the intensity of environmental regulation of neighboring provinces was at a high level, local province was found to have an at least 50% probability of maintaining the intensity of environmental regulation. In addition, the value on the diagonal is higher than that on the non-diagonal, which further indicates that the environmental regulation intensity of each province still has the characteristics of maintaining the stability of the original state grade under the influence of the environmental regulation intensity of neighboring provinces.

(2)     When the intensity of environmental regulation in neighboring provinces was at the medium-low level, the probability of local environmental regulation improving from low level to medium-low level was found to be 6.7%, and the probability of local environmental regulation decreasing from the medium-low level to the low level was 30.4%. When the intensity of environmental regulation in neighboring provinces was at a medium-high level, the probability of local environmental regulation improving from the medium-low level to the medium-high level was found to be 0, and the probability of local environmental regulation decreasing from the medium-high level to the medium-low level was found to be 22.7%. When the intensity of environmental regulation in neighboring provinces was at a high level, the probability of local environmental regulation improving from medium-high level to high level was found to be 27.8%, and the probability of local environmental regulation decreasing from the high level to the medium-high level was found to be 20.5%. This shows that when the intensity of environmental regulation in neighboring provinces is clear, the possibility of downward transfer of local environmental regulation intensity is greater than that of upward transfer, except when the intensity of environmental regulation in neighboring provinces is at a high level.

(3)     When the intensity of environmental regulation in neighboring provinces was at a low level, the probability of local environmental regulation improving from the low level to the medium-low level was found to be 12.5%. When the intensity of environmental regulation in neighboring provinces was at a medium-low level, the probability of local environmental regulation increasing from the medium-low level to the medium-high level was found to be 21.3%. When the intensity of environmental regulation in neighboring provinces was at the medium-high level, the probability of local environmental regulation improving from the medium-high level to the high level was found to be 27.8%. This shows that with the gradual improvement of environmental regulation intensity in neighboring provinces, the intensity of local environmental regulation will also increase.

## 6. Summary and Policy Recommendations

### *6.1. Summary*

In this paper, we mainly discuss three issues. First, we calculated the Theil index in order to analyze whether there are spatial differences in the intensity of environmental regulations among different regions in China. It was found that the spatial difference of environmental regulation intensity in China has greatly fluctuated and intra-interval differences account for more than 85% of the overall differences, indicating that intra-interval differences is are main reason for the spatial difference of environmental regulation intensity in China. Then, we further compared the intensity of environmental regulation in different regions and provinces in order to analyze the inter-regional and intra-regional

differences of environmental regulation intensity in China. It was found that the inter-regional differences of environmental regulation intensity presented a decreasing trend of "west–central–northeast–east". We further analyzed the reasons for inter-regional differences in environmental regulation intensity in China based on the geographical location, environmental pollution status, economic development speed, and urbanization degree of the four regions; it was found that the intra-regional differences of environmental regulation intensity presented a decreasing trend of "west–east–central–northeast". We then analyzed the reasons for the intra-regional differences in environmental regulation intensity in China from the aspects of economic development level, industrial structure, urbanization level, and education level.

Second, we calculated the Global Moran's I index to verify the spatial correlation of China's environmental regulation. That is, given that there are spatial differences in the intensity of environmental regulation in China, we investigated whether the environmental regulation intensity of a province affects the formulation of environmental regulation policies in other neighboring areas and whether the formulation of provincial environmental regulation policies is affected by the intensity of environmental regulation of neighboring enterprises. It was found that there was a significant spatial correlation between the environmental regulation intensity of provinces in 2011, 2012, 2013, 2014, 2015, 2016 and 2017. The Global Moran's I indexes of China's environmental regulation intensity in these 7 years were found to be positive, indicating that an increase in environmental regulation intensity in one province will lead to an increase in environmental regulation intensity in neighboring provinces. Similarly, an increase in environmental regulation intensity in neighboring provinces will also lead to stricter environmental policies in this province.

Third, we used the Markov chain test to analyze the evolution characteristics of environmental regulation intensity shifts in China. That is, since there is a spatial correlation between the intensity of environmental regulation in different regions, we investigated the probability of each province changing the environmental regulation intensity in the process of transferring the environmental regulation intensity due to the influence of the environmental regulation intensity in neighboring provinces. The results of the spatial Markov chain test showed that the probability of local environmental regulation intensity being affected is also different when the intensity of environmental regulation in neighboring provinces is different. The spillover effects of environmental regulation intensity transfer in neighboring provinces were also found to be different.

From the theoretical perspective, the major contribution of this paper is the analysis of China's environmental regulation intensity affected by geographical space from "time and space differences, spatial correlation and spatial spillover effect". We additionally analyzed the theoretical and action mechanisms related to the spatial effect of environmental regulation on the basis of this framework, providing theoretical and logical bases for subsequent research. In order to enrich the research framework of the spatial evolution process and multi-factor driving mechanism of environmental regulation, researchers should fully explore and study the spatial effects of environmental regulation and discuss the regional differences, dynamic evolution rules of spatial patterns, and influencing factors of environmental regulation in China.

From the methodology perspective, the spatial Markov chain method was used to measure the spatial spillover effect of environmental regulation intensity in China. The spatial lag concept was introduced to consider the probability of improving or decreasing the intensity of environmental regulation in neighboring provinces. By analyzing the internal relationship between the intensity of environmental planning and the spatial geographical background, the spatial spillover law of the intensity of environmental regulation in provinces and their internal regions was examined.

### 6.2. Policy Recommendations

A good ecological environment is the fundamental basis for the sustainable development of human society, and environmental quality is a factor that cannot be ignored in the

process of national economic construction. Therefore, to curb environmental deterioration, the government needs to strengthen environmental pollution control, adjust measures to local conditions, and establish an effective regional environmental regulation cooperation mechanisms. In this regard, the authors of this paper propose the following suggestions:

### 6.2.1. Implement Differentiated Regional Environmental Regulations

Local government departments should formulate differentiated regional environmental policies according to different conditions such as local economic development level, geographical location, resource endowment, industrial structure, and the level of opening to the outside world, as well as the spatial distribution and severity of environmental pollution. For economically developed regions, the governments should focus on the advantages of green capital market, emission trading and ecological compensation policies; establish a tripartite environmental governance model featuring government, market, and society; and encourage local enterprises to achieve energy conservation and emission reduction through equipment upgrading and technological innovation. For economically underdeveloped regions, while considering regional economic performance and environmental performance, local governments should constantly optimize their own industrial structure and gradually build a modern industrial system with low energy consumption and less environmental pollution.

### 6.2.2. Strengthen Local Governments' Responsibility for Environmental Protection

Local governments should include environmental protection targets in the responsibilities of leading local government officials during their term of office, and they should improve the environmental protection responsibility system of all functional departments. The governments also need to improve the system of rewards and punishments for environmental protection and the system of accountability for environmental protection, as well as hold leading officials accountable for their negligence in environmental protection. The should gradually build a system in which governments at all levels are responsible for environmental quality; in which enterprises, citizens and social organizations participate; and in which public opinion supervises in order to improve the efficiency of state regulation and control of local environmental protection work.

### 6.2.3. Establish Effective Cooperation Mechanism of Environmental Regulation between Local Governments

Combining geographical location, environmental quality, pollution characteristics and other factors of each region, the state administrative department should scientifically divide regional groups suitable for unified environmental planning and determine environmental quality objectives, pollution prevention and control measures, and key pollution control projects within the region. The governments should further establish a zone of environmental pollution from spreading laws and regulations, a clear regional compensation mechanism in the process of cooperation between environmental regulation and the responsibility mechanism, and a punishment mechanism. The unified supervision, assistance, and evaluation of regional environmental pollution control work will help to improve the efficiency and effect of national environmental pollution control work.

### 6.3. Research Limitations and Future Research Prospects

First, on the spatial scale, the research unit of this paper was 30 provincial administrative units in mainland China. The spatial and temporal distribution characteristics, spatial correlation, and spatial spillover effects of provincial and regional environmental regulation were explored. However, there were limitations in sample selection, e.g., Tibet, Hong Kong, Macao, and Taiwan were not studied due to a lack of data. At present, environmental pollution has become an international governance problem, and local governance approaches are bound to differ. In the future, on the one hand, comparative studies can be conducted in many countries from the macro perspective to analyze the differences

between China and other countries, as well as the spatial differences in environmental regulation intensity between developing and developed countries. On the other hand, the spatial differences of environmental regulation intensity at the city and industry levels can be studied from the micro perspective. There are many prefecture-level cities in China, and each city has its own development model, including "innovative city" and "resource-based city". The environmental pollution status of different cities is different, and there are many differences in environmental governance modes. Therefore, empirical analyses of larger sample sizes can be carried out at the city level.

Second, although the authors of this paper theoretically analyzed various functions and influencing mechanisms of environmental regulations, the reality is very complex, changeable, and may be affected by other factors. In different historical periods or background conditions, the spatial heterogeneity, spatial spillover effect and spatial layout effect of environmental regulation may be different. Future studies can comprehensively analyze the reasons for the spatial differences, spatial correlations, and spatial spillover effects of environmental regulation intensity in provinces and regions from multiple perspectives, including macro factors (such as economic development, industrial structure and urbanization) and micro factors (such as enterprise production mode, investment behavior choice and environmental protection management).

Third, because the means of environmental regulation exist in many forms, there have been many quantitative indicators in past research, though all have certain limitations and shortcomings. The authors of this paper adopted the "environmental regulation inputs" to quantify the environmental regulation intensity, and this indicator is representative to a certain extent but is not perfect.

Future research can be used in more diverse indicators for further exploration. We can try to select indicators from many aspects such as environmental investment, the number of environmental regulations, environmental tax, and public participation, and we can adopt methods such as entropy value method and factor analysis to comprehensively measure the intensity of environmental regulations in various provinces.

**Author Contributions:** Conceptualization, L.F.; methodology, J.S. and W.Z.; software, L.W.; validation, L.F. and W.Z.; formal analysis, J.S. and W.Z.; writing—original draft preparation, L.F. and J.S.; writing—review and editing, L.W. and W.Z. supervision, L.F., J.S. and W.Z. All authors have read and agreed to the published version of the manuscript.

**Funding:** The research work was supported by National Social Science Foundation in China, grant number 18BGL185. The title is "Coupling Mechanism of Environmental Regulation and Corporate Environmental Responsibility and Regional Heterogeneity". The work was funded by Lili Feng.

**Institutional Review Board Statement:** Not applicable.

**Informed Consent Statement:** Not applicable.

**Data Availability Statement:** Not applicable.

**Conflicts of Interest:** The authors declare no conflict of interest.

## Abbreviations

| | |
|---|---|
| CI | Water Conservation Index |
| EKC | Environmental Kuznets Curve |
| EPI | Environmental Performance Index |
| ERI | Environmental Regulation Intensity |
| GDP | Gross Domestic Product |
| GHG | Greenhouse Gas |
| PACE | Pollution Abatement Costs and Expenditures |
| RI | Regulation Index |
| SAR | Special Administrative Region |

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
