# Peer review of "Spatial Correlation and Influencing Factors of Environmental Regulation Intensity in China"

_sustainability, doi:10.3390/su14116504_

Round 1
Reviewer 1 Report
Overall the manuscript is well written, and nicely presented. The following points need to be addressed.
- The abstract should be shortened and reframed
- The introduction is ok but the hypothesis and objectives of the study must be clear.
- The methodology is fairly ok
- Results are ok but the discussion part must be supported with some recently published reports.
- The conclusion is too long it be must be short and client-oriented.
- References should be crossed checked and remove some old references.
Author Response
Dear Editors and Reviewers:
Thank you for your letter and for the reviewers’ comments concerning our manuscript sustainability-1681680 titled “Spatial correlation and influencing factors of environmental regulation intensity in China”. Those comments are all valuable and very helpful for revising and improving our paper, as well as the important guiding significance to our researches. We have studied comments carefully and have made correction which we hope meet with approval.
The main modifications in the article and the replies to the suggestions put forward by reviewers are in Word file.
We tried our best to improve the manuscript and made some changes in the manuscript. These changes will not influence the content and framework of the paper. We appreciate for Editors and Reviewers’ warm work earnestly, and hope that the correction will meet with approval.
Once again, thank you very much for your comments and suggestions.
Yours sincerely,
Wenjun Zhou

Reviewer 2 Report
Dear Authors,
I would like to congratulate you and your team for doing such a good research work in your submitted paper for publication in this prestigious journal. Topic is very interesting and I liked the topic and I personally like to appreciate your efforts to present your research work in such a nice manner. But before your work will be recommended or will be given any possible acceptance few comments must be incorporated for improving the quality of your work as well as for further publication in this reputed journal. I have the following major observations or queries and comments which may further enhance your piece of work. The authors require to modify the following points in detail.
- In abstract, please bring in your 2-3 special quantitative achievements from the results of this study in context of environment by mixing up the research objectives and problems. Please make your abstract within 250 words only. Also, check spellings for many words, which are wrong and are written in hurry.
- The introduction part is required to add few more sentences to increase the strength of this article and kindly bring in the research problem, objective, novelty and explain it in last paragraph of the section of Introduction.
- Add few more sentences in the very beginning of introduction explaining about your paper’s contribution or attempts for dealing or presenting solutions for a specific problem/s and your special contribution with this research paper.
- Please present the methodology section in a compact graphical format as well.
- Literature review part is very weak kindly revise it.
- Please present your literature review in a concise SmartArt chart format.
- Just after the Methodology, benefits of your research for the socio-economic impact with respect to evaluating its determinant.
- The section of “Results” must explain research problems, solutions and the contribution of your study theoretically with around 500-750 words.
- Please insert graphical presentations for your results.
- Explain why have you deployed this study in a separate section of “Policy suggestions” just before the section of “Conclusions”.
- Please add serial numbers to the headings of sub sections.
- Add three more paragraphs (at least of 250 words excluding the existing one) in the section of conclusions mentioning the limitations of the study and remedies to limitations with achieved objectives after conducting this study.
- Add around 150 words more to the section of conclusions explaining the future scope of your research study, limitations if any faced while conducting your research and the procedure to remove the limitations of research.
- Also, explain all the tables more briefly and the explanations of each table in the sections of “Results”.
- Use of English language is very poor. Revise almost all sentences in the manuscript with appropriate use of grammar, punctuation and speech (active or passive voices).
- Add few more suggested references from this and other journals related to the research work. Also, make sure all your references must be cited in body text as well. For referencing, please use Mendeley. Make your references in APA/MLA style of referencing.
- Your results are doubtful, therefore send the data files and methods do files, so that I shall re-conduct or re-investigate the test and rerun the models.
I found that the literature section is a little weak and it requires more studies to be reviewed therefore I suggest you to include the following work:
https://doi.org/10.1016/j.techfore.2022.121524
https://doi.org/10.1016/j.resourpol.2022.102612
https://doi.org/10.1016/j.energy.2022.123619
https://doi.org/10.1016/j.techfore.2022.121570
I think above all studies will make this study more relevant in bridging the gap with literature.
Looking forward for your revised submission.
Author Response

(The authors gave the same response as above.)

Reviewer 3 Report
General comments:
The authors present the influencing factors of, what they call, environmental regulation intensity (ERI) in China. This topic is very interesting. However, in its current state, the manuscript lacks explanation and homogeneity in terminology, especially the real (not only theoretical) criteria how ERI is determined (see also detailed comments). The authors must be aware that the manuscript might also be read by an interested audience not familiar with their nomenclature. Also, the English language needs to be improved, preferably by contacting a native English speaker; especially, the use (or missing use) of definite and indefinite articles must be improved.
In the introduction, the authors state that China ranked place 120 out of 180 in environmental quality. The authors draw however only weak conclusions how China can improve this situation, stating that "the government's assessment of energy conservation, emission reduction and environmental quality in key development areas should not be ignored". In particular, the authors do not discuss if the results of their study, namely the ERI trend, is already an indication that the environmental situation in China is improving or what is still missing to improve it. In the light of internationally well-known smog situations in Chinese agglomerations which I personally witnessed it would be desirable to learn what measures in detail the Chinese government will undertake in the future to improve the environmental situation.
Detailed comments:
Throughout the paper: all abbreviations must be explained when they first appear. Alternatively, a glossary of abbreviations may be included at the beginning of the manuscript.
Line 40: EPI report: please add reference or web site.
Line 42: replace "as a result" by "already", as the referenced National Congress (in 2012) was much earlier than the EPI report (2020). "CPC" must be defined.
Line 85: PACE: define.
Section 3.1: The explanation how local ERI is measured/obtained needs to be extended. In the literature review in Section 2, the authors list different methods to determine ERI; which ones are used here? How are e.g. RGDP, ES, EL and UL determined? How can the obtained value of ERI be interpreted (lines 142-143 state that the bigger the value, the stronger the local environmental regulation) - are there threshold values to distinguish weak - middle - strong environmental regulation? See also comments to the Results section.
Line 148: 30 provinces: from the list in the footnote on p. 10 (which should be transferred to p. 5), I count 31 provinces; please check.
Line 157: Theil index: give reference.
Lines 162-172, equ. 1-3: Much more explanation is needed. All equations are sums, not differences, as stated in lines 166 and 167. The authors apparently mean "regional differences", but it is unclear how these can be determined from equ. 1-3. Also the statements: "TW is the difference among four regions" and "TB is the difference between the four regions" are equal in meaning and wrong because there are no differences. What are "space observation units"? Are these the provinces? What means "Ti is the proportion of the environmental regulation intensity of observation unit i in China"? How are the Ti determined? The same question is related to the regional T values. To be consistent, Tne and Tc should be used for the northeastern and the central regions. The answers to these questions are essential for the readers' understanding of this investigation.
Lines 179 ff: What is the difference between the Ti in equ. 2-3 and the xi in equ. 4 (ERI of province i)? How are the xi determined?
Table 2, Figure 1: How can the numbers be interpreted? What does a mean value of ERI for 2010 of 1.52 % mean? Why is ERI given in %? In % of what? Harmonize units between Table 2, Figure 1, and text. Best to use values in Fig. 1 throughout.
Figure 2, y-axis: values again in %? Are the values along the dotted line equal to those in Fig. 1?
Lines 282/283 (and probably also elsewhere): explain the differences between inter-regional and intra-regional.
Figure 3: again, it is unclear how, using equ. 1-3, differences can be calculated, as these equations contain only sums (see also comment to lines 162-172).
Table 3: explain Z and P. Moran's I values are generally far from 1.
Lines 383/384: I do not see these numbers in Table 5. Instead, I would see the values 21.7 and 0. Explain better how to interpret Tables 1 and 5.
Author Response

(The authors gave the same response as above.)

Round 2
Reviewer 2 Report
Dear Authors,
I would like to congratulate you and your team for doing such a good research work in your submitted paper. Topic is very interesting and I liked the topic and appreciate your efforts to present your revised research work in such a nice manner. I am satisfied from your efforts you employed in the revision and I found all my suggested comments have been incorporated or addressed perfectly. Therefore, I strongly recommended this article for acceptance for further publication in this reputed journal without any more changes.
Author Response
Dear Reviewer:
Thank you very much for your affirmation of our manuscript and research content. We appreciate for your suggestions to help us complete the revision of the paper. We will continue our efforts in the research field in the future.
Once again, thank you very much for your comments and suggestions.
Yours sincerely,
Wenjun Zhou
Reviewer 3 Report
Please see pdf.

Author Response
Dear Reviewer:
Thank you for your letter and for the reviewers’ comments concerning our manuscript sustainability-1681680 titled “Spatial correlation and influencing factors of environmental regulation intensity in China”. Those comments are all valuable and very helpful for revising and improving our paper, as well as the important guiding significance to our researches. We have studied comments carefully and have made correction which we hope meet with approval.
The main modifications in the article and the replies to the suggestions put forward by reviewers are in Word file. We tried our best to improve the manuscript and made some changes in the manuscript. These changes will not influence the content and framework of the paper.
We appreciate for Editors and Reviewers’ warm work earnestly, and hope that the correction will meet with approval.
Once again, thank you very much for your comments and suggestions.
Yours sincerely,
Wenjun Zhou

Round 3
Reviewer 3 Report
Thank you for considering my suggestions from the second round.
Still a few minor suggestions:
Lines 564-569: Tinter is defined in equ. 2, Tintra is defined in equ. 3 (see lines 388-390). The text must therefore be corrected as follows: in line 565, reference must be made to equ. 3. In line 569, reference must be made to equ. 2.
Lines 926, 933: replace "it can be found" by "it was found"
lines 1067-1068: I suggest to write "and this indicator has a certain representativity, but is not perfect."
Author Response

(The authors gave the same response as above.)
